# Microfluidics-Based Fabrication of Cell-Laden Hydrogel Microfibers for Potential Applications in Tissue Engineering

**DOI:** 10.3390/molecules24081633

**Published:** 2019-04-25

**Authors:** Gen Wang, Luanluan Jia, Fengxuan Han, Jiayuan Wang, Li Yu, Yingkang Yu, Gareth Turnbull, Mingyu Guo, Wenmiao Shu, Bin Li

**Affiliations:** 1College of Chemistry, Chemical Engineering and Material Science, Orthopaedic Institute, Soochow University, Suzhou 215006, Jiangsu, China; wanggenboy@163.com (G.W.); lljia@stu.suda.edu.cn (L.J.); fengxuanhappy@126.com (F.H.); wangjiayuan08@163.com (J.W.); 18770918256@163.com (L.Y.); 20174209295@stu.suda.edu.cn (Y.Y.); guomingyu@suda.edu.cn (M.G.); 2Department of Biomedical Engineering, University of Strathclyde, Glasgow G1 1QE, UK; gareth.turnbull@strath.ac.uk (G.T.); will.shu@strath.ac.uk (W.S.); 3China Orthopaedic Regenerative Medicine Group (CORMed), Hangzhou 310000, Zhejiang, China

**Keywords:** microfluidics, microfibers, cell-laden, tissue engineering

## Abstract

Fibrous hydrogel scaffolds have recently attracted increasing attention for tissue engineering applications. While a number of approaches have been proposed for fabricating microfibers, it remains difficult for current methods to produce materials that meet the essential requirements of being simple, flexible and bio-friendly. It is especially challenging to prepare cell-laden microfibers which have different structures to meet the needs of various applications using a simple device. In this study, we developed a facile two-flow microfluidic system, through which cell-laden hydrogel microfibers with various structures could be easily prepared in one step. Aiming to meet different tissue engineering needs, several types of microfibers with different structures, including single-layer, double-layer and hollow microfibers, have been prepared using an alginate-methacrylated gelatin composite hydrogel by merely changing the inner and outer fluids. Cell-laden single-layer microfibers were obtained by subsequently seeding mouse embryonic osteoblast precursor cells (MC3T3-E1) cells on the surface of the as-prepared microfibers. Cell-laden double-layer and hollow microfibers were prepared by directly encapsulating MC3T3-E1 cells or human umbilical vein endothelial cells (HUVECs) in the cores of microfibers upon their fabrication. Prominent proliferation of cells happened in all cell-laden single-layer, double-layer and hollow microfibers, implying potential applications for them in tissue engineering.

## 1. Introduction

Tissue engineering is an exciting research area that has played a pivotal role in replacement and regeneration of tissues and organs. Biomaterials are a crucial mainstay of tissue engineering to provide a three-dimensional (3D) environment for cell growth [1,2,3]. Various polymer materials have been exploited as scaffolds for potentially mimicking extracellular matrixes [4]. Aliphatic polyesters including poly(glycolic acid) (PGA), poly(lactic acid) (PLA), and copolymers (PLGA) of these materials, which are approved for use in the body by the FDA, are the most widely used synthetic polymers [5,6]. However, they are typically processed under extreme conditions, which make bioactive factor incorporation and cells encapsulation a big challenge. As an alternative, a variety of hydrogels are employed as scaffold materials [7]. Hydrogels (especially naturally derived polymeric hydrogels) have structural similarity to the macromolecular-based components in the body and are considered biocompatible [8]. The microstructure design of the scaffold is critical, but hydrogels face significant challenges in this regard [9].

Fiber scaffolds have attracted attention because of their similar structure to natural extracellular matrices [10,11]. Fiber-shaped 3D cellular constructs are ubiquitous in many natural tissues such as muscles, blood vessels, nerves, and osteon. Fibrous extracellular matrices, including collagen, are vital for maintaining tissue structure. Microfibers as a basic structural unit, can be post-processed into various 3D shaped scaffolds. This has prompted many strategies to be proposed for preparing microfibers, such as melt spinning, electrospinning, wet spinning and microfluidic spinning [12,13,14]. Electrospinning and microfluidic spinning are considered to be suitable methods for preparing hydrogel microfibers. However, electrospinning is not suitable for the preparation of cell-laden microfibers because of the need to apply a high voltage or organic solvents during the preparation process.

Recently, microfluidic spinning has shown extensive potential for preparing cell-laden hydrogel microfibers [15,16,17]. The flexibility of microfluidic technology allows the preparation of multi-structured fibers with hollow channels, multi-layer or grooved designs [18,19,20]. Recent studies have also demonstrated potential biomedical applications including the production of heterogenous 3D tissue constructs [21,22]. However, in order to meet different applications, it is often necessary to design a variety of microfluidic devices. Therefore, in order to make the preparation process more flexible and systematic, a microfluidic system that can realize the preparation of various structural microfibers is needed.

In this study, we will develop a facile two-flow microfluidic system that includes two replaceable fluids and a microfluidic device coaxially assembled from glass capillaries. Through this system, cell-laden microfibers with different structures will be prepared using an Alg-GelMA composite hydrogel obtained by mixing alginate (Alg) with methacrylated gelatin (GelMA). The incorporation of GelMA into alginate could improve the cell adhesion on the alginate hydrogel microfibers. It is worth noting that the GelMA we use can crosslink under blue light, which causes little damage to the encapsulated cells in microfibers. In order to demonstrate that the system can fulfill a variety of applications in tissue engineering, various cell-laden microfibers with different structures have been fabricated (Scheme 1). Single-layer, double-layer and hollow cell-laden hydrogel microfibers can be obtained by merely changing the two fluids. Taking mouse embryonic osteoblast precursor cells (MC3T3-E1) and human umbilical vein endothelial cells (HUVECs) as examples, we will explore the potential applications of cell-laden microfibers in bone and vascular tissue engineering. We believe these microfibers will be used in more aspects of tissue engineering.

## 2. Results and Discussion

### 2.1. Fabrication and Characterizations of Alg-GelMA Composite Hydrogel

In order to prepare cell-laden microfibers, we tried to modify the alginate hydrogel with GelMA. GelMA is made from animal tissue extracts and has considerable biocompatibility. Alginate was chosen to prepare microfibers with various structures because of its advantages of rapid gelation. It was reported that superhydrophilic or superhydrophobic surfaces are not conducive to cell adhesion [23,24]. One reason of insufficient cell-favorable microenvironment for alginate is that alginate has superior hydrophilicity. To resolve the above problem, Alg-GelMA composite hydrogel could be easily formed by mixing alginate and GelMA and subsequently crosslinking through both calcium ion and blue light. The water contact angles of alginate hydrogel, culture dish and different Alg-GelMA composite hydrogels are detected (Figure 1). The water contact angle of pure alginate is less than 10°. The water contact angle of 7.5% GelMA is about 82°. While the water contact angle of Alg-GelMA composite hydrogels is bigger than alginate but smaller than 7.5 GelMA. In particular, the water contact angles of Alg-GelMA (1%, 5%) and Alg-GelMA (1%, 7.5%) composite hydrogels are close to culture dish.

Not surprisingly, GelMA hydrogel membrane could support MC3T3-E1 cell adhesion and proliferation (Appendix A). The cell adhesion and proliferation on alginate hydrogel was bad (Figure 2). Alg-GelMA (1%, 5%) composite hydrogel was chosen to prepare membranes for testing cell compatibility. The results show that the cells adhered well and proliferated on the Alg-GelMA hydrogel (Figure 2). The results of this study show the Alg-GelMA composite hydrogel is a biocompatible hydrogel which can support cell growth.

In addition, the mechanical properties of Alg, GelMA and Alg-GelMA hydrogel were investigated. The results show that the compressive modulus of the Alg-GelMA hydrogel is greatly improved compared to the Alg and GelMA hydrogels (Appendix A).

### 2.2. Fabrication of Cell-laden Single-layer Microfibers

In this study, a simple microfluidic device that allowed two fluids to be injected was employed (Appendix A). Microfibers having different structures could be obtained through the microfluidic device. To fabricate single-layer microfibers, two laminar flows were employed in the microfluidic device, calcium chloride (CaCl_2_) solution as outer layer fluid and Alg-GelMA solution as inner fluid. When these two fluids come into contact, a preliminary cured Alg-GelMA microfiber was obtained by immediate ionic crosslinking between calcium ions and alginate. The microfibers were then subjected to a second cross-linking by irradiation under a blue light for 30 s and collected in a petri dish containing CaCl_2_ solution (Figure 3a). The diameter of this single-layer microfiber is about 300 μm, which could be easily controlled by changing flow rate. To explore the potential applications of this microfiber in tissue engineering, MC3T3-E1 cells and HUVECs were separately seeded on their surface. Because cells tend to adhere on the bottom of the dish, we treated the bottom of the cell culture dish using Pluronic F-127 to prevent the cells from adhering to the bottom of the culture dish. The results show that both HUVEC and MC3T3-E1 cells could adhere and grow well on the surface of the microfibers (Figure 3b–e). It is a challenge for cell adhesion on round and small areas, however, cells showed good adhesion on the microfibers prepared in this study.

### 2.3. Fabrication of Cell-laden Double-layer Microfibers

Many natural tissues are made up of solid fibers, such as muscles and bone. In order to apply cell-laden microfibers to more tissue types, we also designed a double-layer microfiber. MC3T3-E1 cells were dispersed in a neutralized collagen solution and then injected into the microfluidic device as an internal phase fluid. The external phase fluid was Alg-GelMA solution. The results show that MC3T3-E1 cells are successfully encapsulated into the core of double-layer microfibers (Figure 4a). Moreover, the encapsulated cells in this microfibers show proliferation after 3, 6 and 9 days of culture (Figure 4a–c). The results of live/dead staining show that most cells in the core of double-layer microfiber are still alive (Figure 4d–f). We also stained the nucleus and cytoskeleton of cells after 9 days of culture to observe the 3D structure and cell morphology of the cell-laden microfibers though confocal laser scanning microscopy (CLSM). The results show that the cells are closely packed and form a solid cell fiber (Figure 5). The penetration and transport of nutrient and waste is a challenge to restrict the cell growth within the microfibers. However, our prepared double-layer microfibers are not subject to this factor and are suitable for cell-laden and support cell growth.

### 2.4. Fabrication and Characterization of Cell-laden Hollow Microfibers

Vessel-like structures are one of the important parts in body. For example, blood vessels, as an important place for blood flow, material exchange between blood and tissue, are receiving more and more attention in the field of tissue engineering. Vascularization plays an important role in the process of 3D engineering tissue implantation and reconstruction [25,26]. To precisely mimic tubular microfiber in natural tissues, we fabricated hollow Alg-GelMA microfibers using the same device for preparing the above microfibers. Two solutions were introduced to the device through the respective inlets, where Alg-GelMA solution was introduced to the outer channel to form individual layers, and 0.5% (*w*/*v*) CaCl_2_ solution was pumped into the inner channel to form the hollow lumen. The outlet of the microfluidic chip was immersed in 1.5% (*w*/*v*) alginate solution in order to collect the microfibers. Various hollow Alg-GelMA microfibers with different inner diameter can be obtained by adjusting the flow rates of inner and outer fluids (Q_in_, Q_out_) (Figure 6). In order to further characterize the hollow structure of the microfibers, the cross section of microfibers is shown in SEM images (Figure 7). The microfibers show obvious cavity structure, the inner diamter and wall thickness of hollow mircofibers are also different. In addition, the perfusion of hollow microfibers was tested and has been supplemented in Appendix A. This provided technical support for the study of blood vessel.

To further explore the applications of hollow fibers in tissue engineering, we prepared HUVECs-laden hollow microfibers to simulate blood vessels. HUVECs were dispersed into the complete medium containing CaCl_2_ and then pumped into the interior of hollow Alg-GelMA microfibers along with the inner flow. The HUVECs are loaded in the hollow channel of the microfibers, and thereby form an endothelium layer (Figure 8a,b), which could simulate the lumen of blood vessels. Live-dead staining results demonstrate the viability of cells layer is higher than 99%. From CLSM images, it could be seen that the HUVEC cell layer formed a similar lumen structure at day 7 (Figure 8d).

### 2.5. Higher-order Assembly Using Hydrogel Microfibers

Microfibers can be assembled into different 3D structures because they are long, thin and flexible [27]. However, hydrogel microfibers are difficult to be assembled due to their insufficient mechanical strength. The microfibers prepared in this study using Alg-GelMA composite hydrogel could be superimposes to construct different structures. In order to prove that these Alg-GelMA microfibers have great potential in creating 3D biomimetic tissue, we have assembled the microfibers by weaving. Several complex 3D architectures are created using microfibers containing fluorescent spheres, including grid structure, warps consisting of three or five microfibers (Figure 9).

## 3. Materials and Methods

### 3.1. Materials

Sodium alginate powder (80~120 cP, Wako Pure Chemical Industries, Osaka, Japan) was dissolved in deionized water at 1% (*w*/*v*). Gelatin methacrylate (GelMA), photoinitiator (lithium phenyl-2,4,6-trimethylbenzoylphosphinate, LAP) and blue light source (3 W, 405 nm) were purchased from Suzhou Intelligent Manufacturing Research Institute (Suzhou, China). GelMA was dissolved in deionized water at 2.5% (*w*/*v*), 5% (*w*/*v*) and 7.5% (*w*/*v*), and the final concentration of photoiniator was 0.25% (*w*/*v*), 0.5% (*w*/*v*) and 0.75% (*w*/*v*), respectively. Sodium alginate was dissolved in 2.5% GelMA solution at a concentration of 1% to obtain a mixed solution, which was named Alg-GelMA (1%, 2.5%) solution. Similarly, we can prepare Alg-GelMA (1%, 5%) and Alg-GelMA (1%, 7.5%) solution. Corresponding composite hydrogels can be fabricated after curing of these mixed solutions. Calcium chloride (CaCl_2_) powder (Sinopharm Chemical Reagent Co., Ltd., Shanghai, China) was dissolved in deionized water at 1.5% (*w*/*v*). Collagen (CellmatrixTM Type I-A) was purchased from Nitta Gelatin Inc. (Osaka, Japan). Pluronic F-127 powder (Sigma-Aldrich, Shanghai, China) was dissolved in deionized water at 10% (*w*/*v*). Fluorescence microspheres were purchased from Aladdin Industrial Corporation (Shanghai, China). All chemicals were used without further purification.

### 3.2. Fabrication of Hydrogel Membranes

To prepare Alg membranes, 200 μL sodium alginate (1%) solution was added to the 12-well culture plate and then 1.5% CaCl_2_ solution was added along the wall to cure. To prepare GelMA membranes, 200 μL GelMA (5%) solution was added to the 12-well culture plate and cross-linked by irradiation under a blue light for 30 s. To prepare Alg-GelMA membranes, 200 μL Alg-GelMA (1%, 5%) solution was added to the 12-well culture plate and cross-linked by irradiation under a blue light for 30 s. Then 1.5% CaCl_2_ solution was added along the wall for second cross-linking.

### 3.3. Mechanical Performance Test of Hydrogel

Alg (1%), GelMA (5%) and Alg-GelMA (1%, 5%) solutions were used to prepare hydrogel cylinders with a diameter of 4.5 mm and a height of 4 mm. Due to the large volume of the cylinder, the photocrosslinking time was set to one minute. The hydrogel cylinders were subjected to a compression test using a mechanical testing machine (HY-0580, Shanghai Heng Wing Precision Instrument Co., Ltd., Shanghai, China) and recorded a stress-strain curve of 30% deformation, wherein the slope of 0–10% deformation was recorded as the compressive modulus. The rate for these compression tests were 2 mm min^−1^. The experiments were performed in triplicates unless otherwise specified

### 3.4. Microfluidics

The microfluidic chip was composed of a set of glass capillaries with different shapes and glass slides. Two cylindrical capillaries (inner/outer diameter: 0.6/1.0 mm) were coaxially assembled inside a square capillary on the slide (Appendix A, Appendix A). The inner dimension of the square glass capillary tube was 1.0 mm. A cylindrical capillary was tapered to the desired orifice (~300 μm) by a micropipette puller (Narishige, Tokyo, Japan) for injection channel. Another one was cut to fixed length (~45 mm) and used as a collection channel. A transparent epoxy adhesive was used to seal the tubes where necessary.

### 3.5. Morphology Characterization and Perfusion Test of Hollow Microfibers

SEM was used to display the morphology of hollow microfibers. The prepared hollow microfibers were washed once in water to remove calcium chloride, and then immersed in deionized water for rapid freezing with liquid nitrogen. Finally, the frozen samples were dehydrated in a freeze dryer.

A needle (32G, inner diameter: 110 μm, outer diameter: 230 μm) was bent into a right angle and inserted into the prepared hollow microfibers. A transparent epoxy adhesive was used to cure the interface between the hollow microfiber and the needle. The needle was connected to a syringe containing water in which red fluorescent microspheres are dispersed by using a plastic tube. The syringe was controlled by a syringe pump. Fluorescence microspheres were purchased from Aladdin Industrial Corporation. A stereomicroscope (OMT-2000D, Oumit Optoelectronics Technology Co., Ltd., Suzhou, China) was used to record the perfusion process after the syringe pump was activated.

### 3.6. Cell Culture

HUVECs were purchased from Procell Life Science & Technology Co., Ltd. (Wuhan, China). The HUVECs were incubated in the incubator at 37 °C with 5% CO_2_ and maintained with Ham’s F-12K medium, supplemented with 0.1 mg mL^−1^ heparin, 0.05 mg mL^−1^ ECGs, 10% fetal bovine serum and 1% penicillin/streptomycin. MC3T3-E1 cells were purchased from Shanghai Zhong Qiao Xin Zhou Biotechnology Co., Ltd. (Shanghai, China) and maintained in α-MEM (Hyclone, Logan, UT, USA), supplemented with 10% fetal bovine serum and 1% penicillin/streptomycin.

### 3.7. Fabrication of Cell-laden Microfibers

To fabricate single-layer microfibers, the inner fluid was 1% (*w*/*v*) alginate and 5% (*w*/*v*) GelMA solution, and the outer layer fluid was 1.5% (*w*/*v*) CaCl_2_ solution. All fluids were pumped into the capillary microfluidic chip by syringe pumps (TJ-3A, LongerPump, Baoding, China). A single-layer preliminary cured Alg-GelMA microfiber was obtained by immediate ionic crosslinking between calcium ions and alginate. The microfiber was then subjected to a second cross-linking by irradiation under a blue light for 30 s and collected in a petri dish containing CaCl_2_ solution. The microfiber was transferred to another dish filled with minimal medium for cleaning to remove residual calcium chloride and photoinitiator. Pluronic F-127 solution (10%) was added to the cell culture plate and aspirated after 24 h. The cell culture plate treated with Pluronic F-127 was air-dried and ultraviolet (UV) light sterilized for use. The single-layer microfiber was moved into the cell culture plate treated with Pluronic F-127, and incubated in the incubator after cells were added.

To fabricate cell-laden double-layer microfibers, collagen pre-gel containing MC3T3-E1 cells was selected as inner fluid, and the outer fluid was Alg-GelMA solution. To fabricate hollow microfibers, Alg-GelMA solution (outer phase) and the medium containing HUVECs and 0.5% (*w*/*v*) CaCl_2_ (inner phase) were pumped into the microfluidic chip. It should be noted that the outlet of the device needs to be immersed in the 1.5% (*w*/*v*) solution for curing. The cell-laden microfibers were then subjected to a second cross-linking by irradiation under a blue light for 30 s and collected in a petri dish containing CaCl_2_ solution. At last, the microfibers were moved into another dish filled with minimal medium for cleaning and collected into cell culture plates containing complete medium and then incubated in the incubator. For all the experiments, the starting flow rates were set as 1 mL h^−1^ and solutions such as Alg-GelMA and CaCl_2_ were filtered using a sterile filter (0.22 μm) to sterilize.

### 3.8. Cell Characterizations

Optical images of cell-laden microfibers were obtained by an inverted microscope (ECLIPSE Ts2, Nikon, Tokyo, Japan). Cells seeded in microfibers were stained with LIVE/DEAD™ Viability/Cytotoxicity Kit (Invitrogen by Thermo Fisher Scientific, Eugene, OR, USA) to investigate the viability. Cytoskeleton and nuclei were stained with TRITC phalloidin and DAPI, respectively, according to the manufacturer’s instructions. Fluorescence photographs of cells-laden microfibers were taken by a fluorescence microscope (EVOS f1, Mill Creek, WA, USA). Fluorescence photographs of cross section of the cell-laden hollow microfibers were taken by a Laser Scanning Confocal Microscope (Zeiss LSM710/780, Oberkochen, Germany).

## 4. Conclusions

In summary, we have prepared several cell-laden Alg-GelMA hydrogel microfibers with single-layer, double-layer and hollow structures using the same two-flow microfluidic system. All preparation processes were completed in one step without changing the device. The single-layer Alg-GelMA microfibers supported cell adhesion and growth on their surface, while double-layer and hollow microfibers could directly encapsulate different types of cells within them, and supported cell proliferation within microfibers. The hollow microfibers also show good perfusion performance. Moreover, the prepared hydrogel microfibers could be easily assembled to construct various structures. Therefore, we believe that this flexible two-flow microfluidic system may have potential applications in tissue engineering.

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
