# Peer review of "Microfluidics-Based Fabrication of Cell-Laden Hydrogel Microfibers for Potential Applications in Tissue Engineering"

_molecules, 2019, doi:10.3390/molecules24081633_

Round 1

Reviewer 1 Report

The authors have presented a thorough investigation on the use of a simple microfluidic device to generate a range of cell-laden hydrogel fibers. Indeed, the use of a two-flow microfluidic device to generate hollow, single-layer, and double-layer microfibers has been well-documented in recent years for a range of materials, including alginate, PLL, PLGA, chitosan, and collagen. To date, there is little evidence of the combination of an ionic-crosslinking methodology with a second (UV) crosslinking step, as is widely used for 3D printing of gelatin methacrylate matrices.

Please clarify the nomenclature used throughout regarding GelMA (1%, 5%) in the experimental section.

The authors need to provide significantly more information regarding the impact on cell viability of immersing the cells in high concentrations of calcium salt (necessary for alginate crosslinking), and the subsequent effect of UV irradiation (albeit for 30s in this instance).

What is the final concentration of photoiniator in the cocktails with respect to GelMA? A solution is made in water at 1.5 %w/v but further details are not forthcoming. Does this mean that after such a short irradiation time that residual photoinitator remains? Does subsequent irradiation, as in the case of confocal microscopy result in any further perturbation of the cells? These details not only influence reproducibility pof these results, but can have a significant implication for cell-viability.

Please provide additional information regarding Figure 2-is the cell culturing performed on flat hydrogel films, how were these produced, polymerized, rinsed etc?

 The authors include information regarding:

Collagen (CellmatrixTM Type I-A) was purchased from Nitta Gelatin Inc. (Japan). Pluronic F-127 powder (Sigma-Aldrich, Shanghai, China) was dissolved in deionized water at 10 %(w/v).

But I cannot find further mention of these materials.

Significant need for English-language proofing need throughout. E.g

Line 47-48: Fibrous scaffolds have attracted attention because have similar structure with nature extracellular matrix

Line 58: because that it is flexible and harmless to cells.

Line 184 : warps of three and five microfibers

Author Response

1. Comment: Please clarify the nomenclature used throughout regarding GelMA (1%, 5%) in the experimental section. Response: Thanks for your good suggestions. We have already described the nomenclature for GelMA (1%, 5%) in the experimental section of the revised manuscript. 2. Comment: The authors need to provide significantly more information regarding the impact on cell viability of immersing the cells in high concentrations of calcium salt (necessary for alginate crosslinking), and the subsequent effect of UV irradiation (albeit for 30s in this instance). Response: Thanks for your good suggestions. During the preparation of the cell-laden microfibers, the cells were immersed in a 1.5% calcium chloride solution only for a short period of time (not more than 2 minutes). After the preparation is completed, the microfibers were moved into another dish filled with minimal medium for cleaning and collected into cell culture plates containing complete medium. The final calcium ion concentration is very low during cell culture. In addition, the curing of GelMA in this study was done by blue light irradiation. Blue light is one of the visible light and has very little damage to cells. And the results of live-dead staining in this study also show that the cells survived very well in the microfibers. 3. Comment: What is the final concentration of photoiniator in the cocktails with respect to GelMA? A solution is made in water at 1.5 % w/v but further details are not forthcoming. Does this mean that after such a short irradiation time that residual photoinitator remains? Does subsequent irradiation, as in the case of confocal microscopy result in any further perturbation of the cells? These details not only influence reproducibility pof these results, but can have a significant implication for cell-viability. Response: Thanks for your good suggestions. GelMA was dissolved in deionized water at 2.5% (w/v), 5% (w/v) and 7.5% (w/v), and the final concentration of photoinitator was 0.25% (w/v), 0.5% (w/v) and 0.75 % (w/v), respectively. The Alg-GelMA microfibers we prepared were very small in size (only about 300 μm). The crosslinking of GelMA is relatively sufficient under 30 seconds of blue light irradiation. Indeed, it is still possible that residual photoinitator remains. Therefore, the microfibers were transferred to the dish filled with minimal medium for cleaning to remove residual photoinitiator. The results of live-dead staining in this study also show that the cells in Alg-GelMA microfibers had good viability. In this work, a 1.5% calcium chloride solution was used to cure the outer layer of the microfibers. In addition, the cells were not irradiated with laser light during the culture and photographed under a laser confocal microscope after live-dead staining. Therefore, the laser does not affect the results of the cell viability test. 4. Comment: Please provide additional information regarding Figure 2-is the cell culturing performed on flat hydrogel films, how were these produced, polymerized, rinsed etc? Response: Thanks for your valuable suggestion. The preparation process of the hydrogel membranes has been supplemented in the revised manuscript. 5. Comment: The authors include information regarding: Collagen (CellmatrixTM Type I-A) was purchased from Nitta Gelatin Inc. (Japan). Pluronic F-127 powder (Sigma-Aldrich, Shanghai, China) was dissolved in deionized water at 10 %(w/v). But I cannot find further mention of these materials. Response: Thanks for your comments. Collagen (CellmatrixTM Type I-A) is used to prepare cell-laden double-layer microfibers, acts as an inner layer to support cell growth. Cells are not easy to adhere on the surface of floating micro-sized materials because cells tend to adhere on the bottom of the dish. To resolve this problem, we treat the bottom of the cell culture dish using Pluronic F-127 to prevent the cells from adhering to the bottom of the culture dish. This content has been supplemented in the revised manuscript. 6. Comment: Significant need for English-language proofing need throughout. E.g Line 47-48: Fibrous scaffolds have attracted attention because have similar structure with nature extracellular matrix Line 58: because that it is flexible and harmless to cells. Line 184 : warps of three and five microfibers Response: Thanks for your valuable comments. We have carefully proofread the language again and made changes accordingly.

Reviewer 2 Report

In the paper, the authors successfully synthesized several type fibers such as single-layer, double-layer and hollow microfibers using Alg-GelMA hydrogel by changing the inner and outer fluids.  The diameter of the fibers can be controlled by changing flow rate.  The cells can adhere to the fibers and proliferated on the fibers.  The authors fully investigated the fibers, such as structure, biocompatibility, cell adhesion property, and processability.  Thus, this work is publishable in Molecules after consideration of the comments below.

Further Comments:

1.         The authors should add the water contact angle of GelMA (7.5%) to Fig. 1

2.         The authors should investigate the cell culture of MC3T3-E1 cells on GelMA (5%) and add to Fig. 2.

3.         The authors should investigate mechanical properties of fibers.  The authors wrote “However, hydrogel microfibers face difficulties in assembly due to their insufficient mechanical strength” (section 2.5). Are the mechanical properties of the Alg-GelMA gel higher than that of hydrogel such as Alg and gelatin gel?

Author Response

1.      Comment: The authors should add the water contact angle of GelMA (7.5%) to Fig. 1

Response: Thank you for the suggestion. The water contact angle of GelMA (7.5%) has been supplemented in Fig. 1 of revised manuscript. We have also added the description in the manuscript.

2.      Comment: The authors should investigate the cell culture of MC3T3-E1 cells on GelMA (5%) and add to Fig. 2.

Response: Thank you for the valuable suggestions. The cell culture of MC3T3-E1 cells on GelMA (5%) has been investigated in the revised manuscript. Not surprisingly, the cells adhered and proliferated well on the surface of the GelMA hydrogel (Figure S1). We have also supplemented the description in the manuscript.

3.      Comment: The authors should investigate mechanical properties of fibers.  The authors wrote “However, hydrogel microfibers face difficulties in assembly due to their insufficient mechanical strength” (section 2.5). Are the mechanical properties of the Alg-GelMA gel higher than that of hydrogel such as Alg and gelatin gel?

Response: Thanks for the critical comment. We have studied the mechanical properties of Alg, GelMA and Alg-GelMA hydrogel in the revised manuscript. The data has been supplemented into Supporting Information (Figure S2). The results show that the compressive modulus of the Alg-GelMA hydrogel is greatly improved compared to the Alg and GelMA hydrogels. We have also supplemented the description in the manuscript.
